# Training a Tokenizer for Free with Private Federated Learning

**Eugene Bagdasaryan**[*]
Cornell Tech
eugene@cs.cornell.edu

**Congzheng Song** and **Rogier van Dalen** and **Matt Seigel** and **Áine Cahill**
Apple
{csong4,rogier_vandalen,mseigel,aine_cahill}@apple.com

## Abstract

Federated learning with differential privacy, i.e. private federated learning (PFL), makes it possible to train models on private data distributed across users' devices without harming privacy. PFL is efficient for models, such as neural networks, that have a fixed number of parameters, and thus a fixed-dimensional gradient vector. Such models include neural-net language models, but not tokenizers, the topic of this work. Training a tokenizer requires frequencies of words from an unlimited vocabulary, and existing methods for finding an unlimited vocabulary need a separate privacy budget.

A workaround is to train the tokenizer on publicly available data. However, in this paper we first show that a tokenizer trained on mismatched data results in worse model performance compared to a privacy-violating "oracle" tokenizer that accesses user data, with perplexity increasing by 20 %. We also show that sub-word tokenizers are better suited to the federated context than word-level ones, since they can encode new words, though with more tokens per word.

Second, we propose a novel method to obtain a tokenizer without using any additional privacy budget. During private federated learning of the language model, we sample from the model, train a new tokenizer on the sampled sequences, and update the model embeddings. We then continue private federated learning, and obtain performance within 1 % of the "oracle" tokenizer. We show that, since this process trains the tokenizer on the server using data for which the privacy loss has already been accounted for, our method spends no additional privacy budget.

## 1 Introduction

Learning a language model (LM) requires text data that in many situations is private, resides on people's devices, and should stay there. In federated

learning (McMahan et al., 2017), a central server learns a model by receiving statistics, like parameter updates, from many devices. Though devices send only statistics and not the raw data, federated learning by itself can leak information about the data (Shokri et al., 2017; Song et al., 2017). Private federated learning (PFL) (McMahan et al., 2018; Geyer et al., 2017) uses differential privacy (Dwork et al., 2006, 2014) to mitigate the privacy leaks by limiting the user's impact on the final model.

It is known how to train neural-net language models using PFL (McMahan et al., 2018). However, an important part of language modeling is tokenization: turning a text into a sequence of symbols from a fixed-size symbol set. To obtain a tokenizer, published research on private federated learning of language models uses either of two approaches, neither of which are satisfactory. One approach is to train the tokenizer on user data directly. The commonly-used LEAF dataset (Caldas et al., 2018) and works relying on it (Li et al., 2021; Hu et al., 2021; Yu et al., 2020) assume access to the training data to create the tokenizer. This is not relevant to real-world use cases and undermines user privacy. The other approach is to use public data to obtain the tokenizer (McMahan et al., 2018). This is sensible from a privacy perspective, but as we show the resulting distribution mismatch harms performance, resulting in 10%-20% drop compared to using an "oracle" tokenizer trained directly on users' private data.

There are two common types of tokenization, which are affected by mismatched distributions in different ways: word and sub-word tokenization. Figure 1 illustrates these. A word-level tokenizer produces a symbol for each word, and assigns an out-of-vocabulary token (OOV) to any unseen word. Text from mismatched distributions will generally contain unseen words, which means the correct word cannot be predicted, and the context becomes less meaningful when predicting the

---

[*]Work done during an internship at Apple.

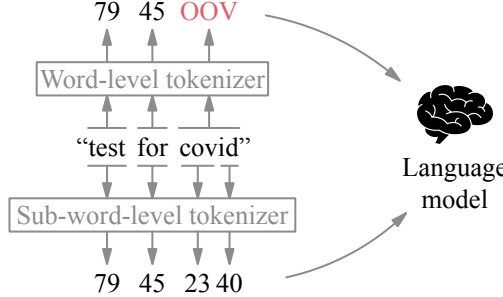

Figure 1: Word-level and sub-word-level tokenization. A word-level tokenizer can generate an "out-of-vocabulary" (OOV) symbol, which it is hard for a language model to use.

next word. Sub-word tokenization, on the other hand, splits some words into multiple smaller tokens. This type of tokenization is generally chosen to minimize the average number of tokens per word on training data. Current centrally trained models use sub-word tokenization such as Byte-Pair Encoding (Sennrich et al., 2016), SentencePiece (Kudo and Richardson, 2018), or WordPieces (Schuster and Nakajima, 2012). Nevertheless, mismatched tokenizations in sub-word methods cause an increase in the number of tokens per word, and thus decrease the amount of context the model can use to predict the distribution of the next word.

In this work we present a general framework to approach training language models in private federated learning by including tokenization as part of the training pipeline. Our contributions are: (1) we uncover the performance gaps when the models use the tokenizer obtained from a different distribution vs the tokenizer obtained from the underlying distribution. For word-level tokenization we show that a tokenizer trained on public data reduces the next-word prediction accuracy of 10–20 % compared to a tokenizer estimated on user data. (2) We demonstrate significant benefits of switching tokenizers from word to sub-word level, thus eliminating the out-of-vocabulary problem. (3) We propose a new method that samples data from an existing model, e.g. from the prior PFL run, and uses that data to initialize a new tokenizer. Our approach can update the tokenizer between iterations of the same PFL run by modifying model embeddings with new tokenizations and significantly boosting performance. Crucially, since the additional processing is done entirely on the server, training the tokenizer with our approach does not use any additional privacy budget.

## 2 Private federated learning

Machine-learned models work best if they are trained on the correct distribution of the data, in this paper text data. In many scenarios text data is private and contained on people's devices, and should stay there. To train a global model without harming privacy, we use federated learning (McMahan et al., 2017) with differential privacy (Dwork et al., 2006, 2014).

Federated learning involves devices sending not the data, but statistics, e.g. model gradients, computed on that data. To train neural networks, the standard algorithm is *federated averaging* (McMahan et al., 2017). At each iteration $t$, the server randomly selects a subset of $m$ participants $S_m$ and distributes the current global model $M^t$. Each participant takes a number of gradient steps to train on their private data and submits the sum $G_i^t$ of the gradients to the server. The server takes a step (with step size $\eta$) in the direction of the average gradient to create the new global model:

$$M^{t+1} = M^t + \frac{\eta}{m} \sum_{i=1}^{m} G_i^t \qquad (1)$$

### 2.1 Federated Learning with Differential Privacy

The global model $M^{t+1}$ might still reveal private information including user participation in training (Shokri et al., 2017; Song et al., 2017; Melis et al., 2019). To mitigate this threat, we can combine federated learning with differential privacy (DP) (Dwork et al., 2006, 2014), to give *private federate learning* (McMahan et al., 2018). Differential privacy gives a strong guarantee: it limits the advantage that a computationally unconstrained adversary has in inferring whether an individual's data is contained in the data set that the statistics are computed from. $(\epsilon, \delta)$-differential privacy parametrizes this advantage by $\epsilon$ (the maximum privacy loss) and $\delta$ (a slack term). The common mechanism to provide differential privacy in a federated learning setting is the Gaussian mechanism that uses the *moments accountant* (Abadi et al., 2016). Each participant clips its gradients to a norm $S$, i.e., multiplied by $\min(1, S/\|G^t\|_2)$, to bound the sum's sensitivity to any individual's data. Second, Gaussian noise $\mathcal{N}(0, \sigma^2)$ is added to the sum.[1] How much privacy budget is spent in one iteration depends on the

---

[1] In practice, a technique like secure aggregation (Bonawitz et al., 2017) can allow central DP on a sum without having to trust the server (Goryczka and Xiong, 2015).

variance $\sigma^2$ relative to the magnitude of individual updates, the total population, and the number of contributions (for more details, see McMahan et al., 2018; Balle et al., 2018). The moments accountant keeps track of this in terms of the *Rényi differential privacy* (Mironov, 2017). What is learned in one iteration is allowed to affect the query in the next iteration, and this increases the budget (in terms of Rényi DP) merely linearly. This is called *adaptive composition*, and it is crucial both to standard private federate learning (where the model changes every iteration as in (1)) and to the method we propose.

## 2.2 Privately finding vocabulary items

Central differential privacy with the Gaussian mechanism and the moments accountant is efficient in terms of utility vs privacy loss, but it does come with restrictions. The sum of individual contributions, which the noise is added to, must be of finite and fixed size. This is not a problem for training neural networks. However, training a tokenizer requires frequencies for an exponential-size set of sequences, as does training a traditional $N$-gram model. Differentially private algorithms to compute histograms over sets of elements (e.g. words) distributed over devices are called "heavy hitters" algorithms (Bassily et al., 2017; Zhu et al., 2020; Apple, 2017). These algorithms require a separate and large privacy budget. In section 5 we will compare with a heavy hitters algorithm.

Another way of finding vocabulary items privately is to train a neural-net generative model. Beaufays et al. (2019) trains a separate, character-level LSTM model to generate the new words. However, the proposed method is only shown to work for discover OOVs in a word-level model and also requires separate training and a privacy budget.

## 3 Tokenization in Language Modeling

A language model is a model that assigns probabilities to sequences of tokens. In this paper, it is always an autoregressive model with parameters $\theta$: $P_\theta(s) = P_\theta(t_2|t_1 = \text{BOS}) \cdot P_\theta(t_3|t_1 = \text{BOS}, t_2) \cdots P_\theta(t_n = \text{EOS}|t_1 = \text{BOS}, \ldots, t_{n-1})$, where each term in this equation is normalized over all possible values of the current token. Local normalization is useful when decoding input, like in speech recognition or a keyboard (Hard et al., 2018). For this paper, we assume that a corpus is segmented into sentences. A tokenizer $\tau$ then con-

verts each sentence $s$ in the dataset into a sequence of $n$ tokens $\tau(s) = [\text{BOS}, t_2, .., t_{n-1}, \text{EOS}]$, which is fed into the language model. There are two types of tokenization, highlighted in Figure 1: word-level and sub-word-level. Using a sub-word tokenizer will be key to the algorithm this paper proposes.

The next section will discuss the two types of tokenizers and their consequences for out-of-vocabulary tokens and the performance of language models based in them. Section 3.2 will discuss the complex topic of how to compare performance across different tokenizations.

### 3.1 Word-level vs sub-word-level tokenization

The type of tokenization that papers about language models in federated learning commonly use is word-level tokenization (McMahan et al., 2017). For a vocabulary of size $N$ the tokenizer assigns a unique token for top-$N$ most popular words in the dataset while other words receive an out-of-vocabulary token OOV, as highlighted in Figure 1. Some papers (e.g. McMahan et al., 2018) build the tokenizer from a publicly available dataset, others including the LEAF benchmark (Caldas et al., 2018) build the tokenizer from users' training data. OOV tokens in the word history make it harder for a language model to predict the next word.

The other type of tokenization is sub-word tokenization, for which there are two popular schemes: byte-pair encoding (BPE) (Sennrich et al., 2016) and WordPieces (Schuster and Nakajima, 2012). We focus on BPE which unlike WordPieces guarantees the absence of OOVs as there exists a token for every byte. However, the number of tokens required to encode each word can change significantly depending on the dataset that the tokenizer was trained on. As highlighted in Figure 1, a tokenizer trained on data from before the COVID-19 pandemic would generate multiple tokens for the word "covid".

Generating longer token sequences makes it harder for the language model to keep track of the context, degrading its performance. Even LSTMs and transformers, which in theory can use arbitrarily long history, have imperfect memory.

### 3.2 Evaluating language models across tokenizations

Comparing language models across tokenizations is a complex problem. For example, when comparing word-level language models using perplexity, often OOVs are ignored which gives an edge to

the language model with more OOVs, which is the opposite of what is desired. The following sections detail the problems when comparing sub-word language models.

### 3.2.1 Comparing word-level with sub-word

Since a word-level language model has a closed vocabulary, it outputs probabilities only on in-vocabulary words, artificially lowering the perplexity of closed-vocabulary LMs, particularly on data with a large number of OOVs. Removing those same words in evaluating a sub-word language model, would disadvantage it.

A better alternative, which this paper will use, is to compare model performance the word-level accuracy. The most accurate way would be to find the word with the highest probability by summing over sequences of tokens. However, we choose a simpler, though less accurate method (similar to Likhomanenko et al., 2019): repeatedly generate the best tokens within each word's bounds and only accept the word as accurate if all generated tokens were correct.

### 3.2.2 Comparing sub-word with sub-word

It is possible to meaningfully compare perplexities of two language models with different sub-word tokenizations (Mielke, 2019). Though the language model assigns probability mass to all token sequences, a single sentence can have multiple corresponding token sequences, only one of which will be chosen by the tokenizer. Some of the probability mass will therefore be lost to never-occurring token sequences. However, it is unfeasible to sum over all token sequences (Likhomanenko et al., 2019).

The danger with comparing perplexities directly is that since models with different tokenizers operate on different sets of tokens the number of tokens needed to encode each sentence is different in general (Mielke, 2019). Nevertheless, note that all models assign a probability to a sentence (with the approximation above). To compute the perplexity in such a way that it can be compared across tokenizers, use the same denominator in computing the perplexity: the number of words in the sentence instead of number of tokens, which depends on the tokenizer. Therefore we define the perplexity as:

$$ppl_{\theta,\tau}(s) = \exp\left(\frac{-\log(P_{\theta,\tau}(s))}{\|s\|_w}\right) \quad (2)$$

where $\|s\|_w$ counts the number of words in the sentence $s$. To generalize from a single sentence

to a dataset, replace $s$ with the concatenation of all sentences in the dataset.

## 4 Learning a Tokenizer with Private Federated Learning

***Problem definition.*** We aim to obtain a tokenizer that works well on users' federated data without compromising user privacy. First, we aim to find the appropriate tokenization scheme, and second, given the tokenization scheme obtain the right approximation of user data to train the tokenizer.

***Setting.*** We focus on a common application of federated learning: training a language model, parameterized by $\theta$, using federated learning with differential privacy. In our setting, each user $u_i$ has a dataset $d_i$ of private texts from a private distribution of user data $\mathcal{D}$. The trained model will be evaluated against a held-out dataset $\mathcal{D}_{test}$, e.g. a mix of all user data, which in practice must be replaced by federated evaluation.

We assume that the central server does not have access to the user data distribution $\mathcal{D}$ and can only approximate it with the publicly available dataset $\mathcal{D}_{pub}$. We assume the public data is some commonly available dataset, such as Wikipedia (Merity et al., 2017). The tokenizer trained on this public data will be $\tau_{pub}$. For comparison we assume the existence of an *oracle* tokenizer $\tau_o$ initialized on users' training data $\mathcal{D}$.

Papers that study language models in federated learning commonly use word-level tokenization. While some papers (e.g. McMahan et al., 2018), build the vocabulary using publicly available dataset, others (e.g. Yu et al., 2020; Caldas et al., 2018) explicitly use the federated training data, even though in real-world scenarios the analogous data would be unavailable and it violates privacy guarantees when used in PFL (Li et al., 2021).

### 4.1 Sampling from a PFL-trained language model

To address the problem of learning a good tokenizer we first propose to use a sub-word tokenizer with an open vocabulary. This allows the language model trained with such a tokenizer to represent any word, if inefficiently. It is then possible to query the language model to find new words as the model can utilize this open vocabulary. This is the core of the Algorithm 1 that this paper introduces.

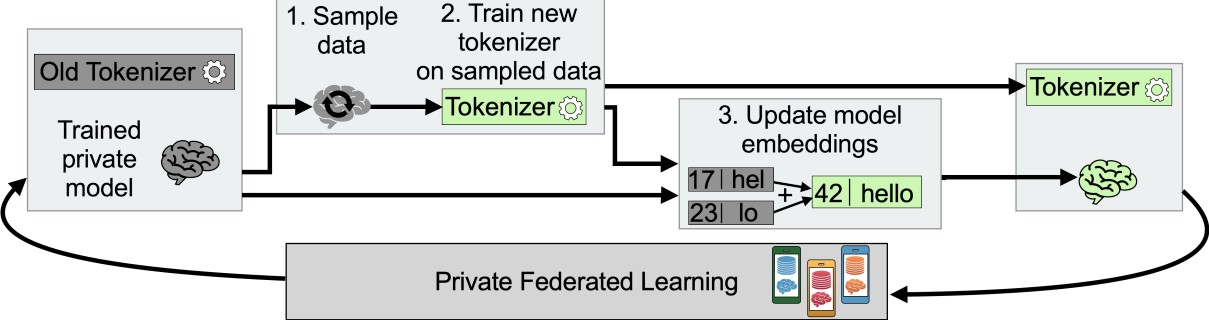

Figure 2: New pipeline for updating the tokenizer through model sampling.

Figure 2 shows the proposed pipeline. A language model is trained with private federated learning. This results (on the left) in a model matched with an old, stale tokenizer. The next block queries the language model to produce a better tokenizer, with a method that section 4.2 will detail. The block after that updates the language model for the new tokenizer, using reasonable guesses for the new parameters. This results in a new LM-tokenizer combination that can be trained further with PFL.

Adaptive composition (see Mironov, 2017) of differential privacy makes it possible to run a server-side process between iterations without spending additional privacy budget. The function UPDATE in Algorithm 1 performs the on-server steps. The following sections will give more detail.

### 4.2 New tokenizer from a trained LM

Training a tokenizer requires text data. Since the raw data is not available, we propose to instead sample from the LM matched with the stale tokenizer, as detailed in Algorithm 1. The SAMPLETOKENS function samples from the language model, drawing sequences of tokens according to the probabilities that the model assigns to them. The SAMPLE function then converts these sequences in the old tokenization into word sequences, by decoding with $\tau_{pub}$. Once a large enough corpus of word-level sentences has been produced, training a tokenizer proceeds as normally (the TRAINTOKENIZER function is not specified).

### 4.3 Adapting the language model to the new tokenizer

After a new tokenizer $\tau$ has been trained, the language model, trained with $\tau_{pub}$, must be updated to work with the new tokenizer. Neural-net language models use an embedding layer to convert the provided tokens into multi-dimensional vectors.

It is the embedding vectors that are most important to modify when changing the tokenization. The rest of the model only consumes the embedding vector. It is not possible to find the optimal parameters without further training of both embeddings and other layers, but we propose an algorithm to find a reasonable starting point, in the function REMAP($\tau, \tau_{pub}$) in Algorithm 1.

REMAP iterates over the tokens from the new tokenizer $\tau$ and creates the mapping from the tokens' embedding in the public tokenizer $\tau_{pub}$ to the new token's embedding. In some cases it is a one-to-one mapping, but when the new token accumulates multiple tokens in $\tau_{pub}$ we split the weight equally between each token.

Once we have the mapping $map$ we modify the embedding layer of the model by performing matrix multiplication, i.e. $\theta.\text{embedding} = map \cdot \theta.\text{embedding}$. The resulting model can accept the tokens from the new tokenizer $\tau$, and can participate in future training in federated learning.

## 5 Experiments

We evaluate our approach by first looking at performance of tokenizers trained on the distributions matched and mismatched to real data, we then test the proposed federated sampling on different datasets for federated learning.

### 5.1 Experimental setup

We use two datasets common in the federated learning literature (Kairouz et al., 2019). While both use English, there is nothing about our experiments that is specific to this language, and multilingual datasets can further benefit from using Sentence-Piece tokenization (Kudo and Richardson, 2018).

- Reddit data – this dataset is taken from the LEAF benchmark (Caldas et al., 2018) and

**Algorithm 1** Model sampling algorithm

**Inputs:** model $\theta$, current sentence $s$, new tokenizer $\tau$, public tokenizer $\tau_{pub}$, size of the sampled dataset corpus_size.

**function** SAMPLETOKENS($\theta, s$)
    $t_{next} \sim_\theta t_k | s$
    **if** $t_{next} = \text{EOS}$ **then**
        **return** $s \mathbin{++} t_{next}$
    **else**
        **return** SAMPLETOKENS($\theta, s \mathbin{++} t_{next}$)

**function** SAMPLE($\theta, \tau$)
    **return** $\tau$.decode(
        SAMPLETOKENS($\theta, [\text{BOS}]$))

**function** REMAP($\tau_{pub}, \tau$)
    map $= \text{zeros}(\tau.\text{size}, \tau_{pub}.\text{size})$
    **for** token, tid $\leftarrow \tau$.vocab **do**
        tokens $= \tau_{pub}$.decode(token)
        **for** token $\leftarrow$ tokens **do**
            $\text{tid}_{pub} = \tau_{pub}$.vocab[token]
            $\text{map}[\text{tid}_{pub}, \text{tid}] = 1/\text{len}(\text{tokens})$
    **return** map

**function** UPDATE($\theta, \tau_{pub}$)
    **while** len(corpus) $<$ corpus_size **do**
        corpus $\leftarrow$ SAMPLE($\theta, \emptyset, l_{max}$)
    $\tau = $ TRAINTOKENIZER(corpus)
    map $=$ REMAP($\tau_{pub}, \tau$)
    $\theta$.embedding $= \text{map} \cdot \theta$.embedding
    **return** $\theta, \tau$

contains over a million users that have multiple posts on the Reddit platform. As proposed by LEAF, we limit each user to contain at most 1600 tokens and use 10 % of users for faster training.

- StackOverflow data – this data is taken from Kaggle (Kaggle, 2021) and processed with the TensorFlow Federated framework. The train split of the dataset contains 342k users and we select at most 1600 tokens per user.

***Model parameters.*** We use an LSTM model with 3 layers, and total parameters of 14M. We also use a Transformer language model (Vaswani et al., 2017) with 6 layers and the same total number of parameters as the LSTM (see Appendix A). Each model is trained from scratch.

***Hyper-parameters.*** We set the privacy budget to $\epsilon = 2$ and $\delta = 10^{-6}$ – a common privacy

regime (Kairouz et al., 2019). For the "heavy hitters" baseline we use local DP with an additional privacy budget of $\epsilon = 8$.[2] The overall population for the moments accountant is assumed to be 10m. We use a cohort size of $20,000$ for each round and train all models for $5,000$ iterations. We use Adam (Kingma and Ba, 2015) for central optimization with learning rate set to 0.5. For the clients we use SGD and train for 1 local epoch with batch size set to 16 and local learning rate set to 0.1, and an $L_2$ clipping bound for DP of 0.5.

***Vocabulary size.*** We assume that the tokenizer has a moderate vocabulary size such as 10,000 tokens (we experiment with larger vocabularies in Appendix A). Smaller vocabularies reduce model size and, therefore, might be better for deployment on devices and communication with the global server.

***Tokenizer details.*** To train an initial tokenizer (on the server) we use a popular and public Wikipedia dataset (Merity et al., 2017). It may seem like the distribution of Wikipedia data is artificially far from the distributions of Reddit and StackOverflow data. However, the server might not have the right prior possibly due to a natural *distribution shift* (Miller et al., 2020) of typed texts (such as an emerging topic of which there were plenty recently).

We use BPE and WordLevel tokenization algorithms from the HuggingFace Tokenizer library (Huggingface, 2021). Each user post is surrounded by special tokens BOS and EOS. We also tried WordPieces tokenization which has slightly better performance than BPE but cannot encode all words and is therefore less applicable in FL.

***Note on splitting data.*** Whereas the original LEAF dataset for Reddit proposes to split each user's data we argue that in real life not every user might have a chance to participate in the training. Therefore, we split users into two distinct training and test sets and evaluate the model on data from the users who have never participated in the training. This results in notably increased test perplexity but provides a clear separation between training and inference modes.

## 5.2 Comparing tokenization schemes

Table 1 summarizes experiments that use different tokenization schemes. We compute statistics on tokenizers: the average share of OOV tokens for the

---

[2] Budgets for local and central privacy are not immediately comparable, but see Feldman et al. (2021).

Table 1: Word accuracy suffers for word-level tokenization that uses mismatched data.

| Type | Data to train $\tau$ | $\tau$ statistics OOV (%) | $\tau$ statistics Tokens per word | Word Accuracy (%) |
|---|---|---|---|---|
| | | *Reddit* | | |
| Word-Level | Wiki | 13.0 | 1.00 | 17.7 |
| Word-Level | Oracle | 5.5 | 1.00 | 24.1 |
| BPE | Wiki | 0.0 | 1.32 | 22.2 |
| BPE | Oracle | 0.0 | 1.22 | 22.5 |
| | | *StackOverflow* | | |
| Word-Level | Wiki | 9.8 | 1.00 | 30.0 |
| Word-Level | Oracle | 2.0 | 1.00 | 33.0 |
| BPE | Wiki | 0.0 | 1.41 | 31.8 |
| BPE | Oracle | 0.0 | 1.24 | 32.4 |

Table 2: Tokenizers initialized on sampled data perform very close to using "oracle" data.

| Type | Data to train $\tau$ | Data KLD | Tokens p/word | LM Acc. (%) | LM Perp. |
|---|---|---|---|---|---|
| | | *Reddit* | | | |
| BPE | Wiki | 0.78 | 1.32 | 22.2 | 276.5 |
| BPE | Oracle | 0 | 1.22 | 22.5 | 256.9 |
| BPE | Heavy hitters[*] | 0.09 | 1.30 | 22.1 | 274.2 |
| BPE | **Sampled** | 0.02 | 1.22 | 22.5 | 257.7 |
| | | *StackOverflow* | | | |
| BPE | Wiki | 1.06 | 1.41 | 31.8 | 124.6 |
| BPE | Oracle | 0 | 1.24 | 32.4 | 108.2 |
| BPE | Heavy hitters[*] | 0.10 | 1.29 | 32.1 | 115.9 |
| BPE | **Sampled** | 0.01 | 1.23 | 32.4 | 108.7 |

[*]The "heavy hitters" algorithm uses local DP and requires additional privacy budget.

word-level scheme and the average number of tokens required to encode one word for the sub-word scheme. To compare the effect of each tokenizer on the PFL-trained model, we report word-level accuracy, for the reasons described in Section 3.2. The "wiki" tokenizers are trained on the Wikipedia data, and the "oracle" tokenizers directly on the training data.

Word-level tokenization provides high word accuracy when it is trained using "oracle" user training data. However, when the word-level has access to only public "wiki" dataset that mismatches user distribution the performance significantly drops: by 26 % for Reddit and 10 % for StackOverflow with a significant increase in out-of-vocabulary share. However, BPE tokenizers that use public data perform more consistently and outperform the word-level models trained on public data, but still require a large number of tokens per each word.

## 5.3 Learning a tokenizer with sampling

A key part of the proposed algorithm is the sampling from a model that uses a public tokenizer $\tau_{pub}$, but is trained with private federated learning and should represent the words in the actual data. The sampling is implemented as in Algorithm 1.

First, Figure 3 shows samples from the language models on the two data sets. Although clearly the samples are less coherent than the underlying data, it seems plausible that the word occurrences match that data.

Second, Table 2 further investigates the properties of the sampled text. The "BPE sample" rows refer to the method proposed in this paper. A language model with the "wiki" tokenizer is trained

with PFL on the first half of the training data. Then samples are drawn from this language model. Then, the language model is trained from scratch on the second half of the training data.

The "BPE Heavy hitters" rows refer to training with a differentially private "heavy hitters" algorithm (Apple, 2017). Each of the population of the users from the first half of the training set contributes three words from the from the Wikipedia dataset, with a local privacy budget of $\epsilon = 8$. Just like for the sampling approach, the language model is then trained from scratch on the second half of the training data.

First, we examine the difference between the real training data and the data used to train the tokenizers. The column "Data KLD" shows the KL divergence from the user "oracle" training data to the sampled data. The KL divergence is computed from the unigram counts, which are relevant for training a tokenizer, over the top 10,000 words

*Reddit*
i would love to know why we may already live in a consolation subreddit and the aforementioned it will almost always be done on the warrior sheet shows from the west . i

*StackOverflow*
json results are : can anyone provide a complete sample response ( lists of descendants list ) to my page depending on future python functions . in web apps that require patient for many

Figure 3: Example of sampling data from the model.

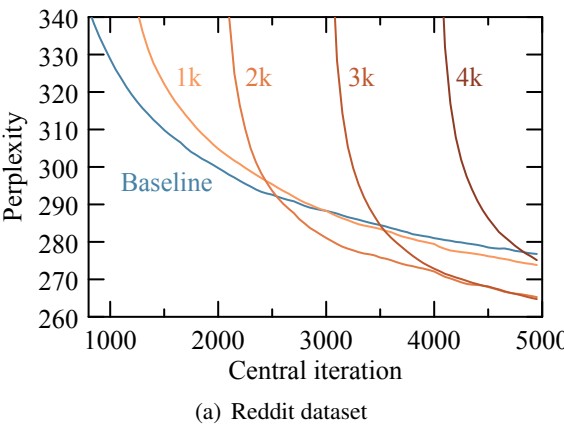

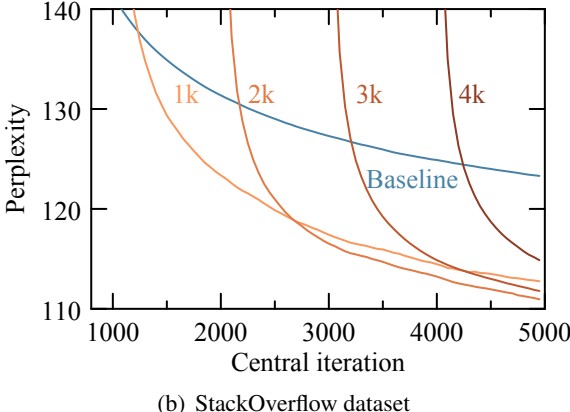

(a) Reddit dataset      (b) StackOverflow dataset

Figure 4: Perplexity for switching the tokenizer at different rounds of federated learning.

from the training data and with add-1 smoothing. The KL divergence to the training data itself, which the oracle tokenizer is trained on, is 0 by definition. The KL divergence between the actual data and the Wikipedia data, on the other hand, is around 1, for both datasets. Both the heavy hitters algorithm and the algorithm we propose in this paper find a distribution close to the real distribution.

For sub-word tokenizers, the number of tokens per word is relevant. Even though they can represent unseen words by multiple tokens, a language model trained on top of that has a harder task given the longer context on average. The oracle tokenizer has the lowest number of tokens per words and the "wiki" tokenizer the highest. The "BPE sample" tokenizer comes very close to the oracle tokenizer.

However, the local-DP heavy hitters experiment shows much smaller gain in performance, i.e. better than "wiki" tokenizer but still worse than our proposed sampling method. Furthermore, it requires a separate privacy budget allocated for the run, while sampling can operate on existing prior model.

### 5.4 Iterative updates

This part implements Algorithm 1 completely. We again initialize the tokenizer on publicly available data. We then train the language model with PFL. At a point during training, we retrain the tokenizer by sampling. Unlike in the previous section, we update the language model by remapping its embedding layer, and continue training. We sample the same data before and after changing the tokenizer.

Figure 4 shows the results for changing tokenizers at different times. The "Baseline" curve represents the model trained using public tokenizer $\tau_{pub}$ from Wikipedia data. Each of the other curves takes the system from the "Baseline" curve at a different iteration. As expected, the initial remapping of the embedding layer is not perfect and needs finetuning. The graph also shows the tradeoff in when to change tokenizers: too early, e.g. after only 1000 iterations, and the tokenizer is not representative enough yet; too late, e.g. after 4000 iterations, and there is not enough time to converge again.

## 6 Conclusion

This paper has proposed a method that allows a tokenizer to be found together with a language model using private federated learning. First, it has shown that a mismatched tokenizer can cause a significant performance degradation. The key to improving this is to use a sub-word tokenizer which allows new words to be represented as a sequence of tokens. Then, a language model trained with PFL can represent the private data. This paper has presented a method to produce a new tokenizer from that model without spending additional privacy budget, and to convert the model to work with the new tokenizer. When this is trained further with private federated learning, it outperforms the language model with the mismatched tokenizer, and gets close to one with the oracle tokenizer.

***Personalization and Fairness.*** The problem of out-of-vocabulary words might be more acute for some users that use unique vocabulary, such as dialect, and impact individual performance. Therefore good tokenizers can benefit personalization in federated models (Li et al., 2021; Yu et al., 2020).

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

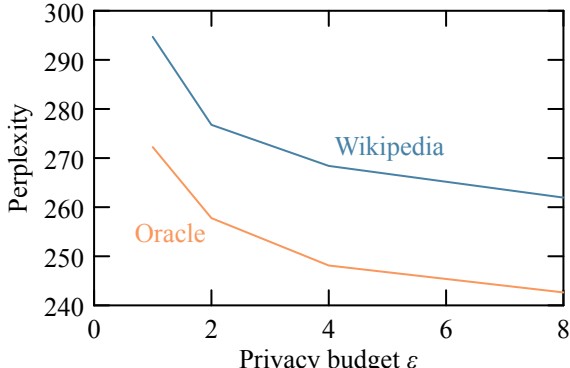

Figure 5: Perplexity trained with different privacy parameter $\epsilon$.

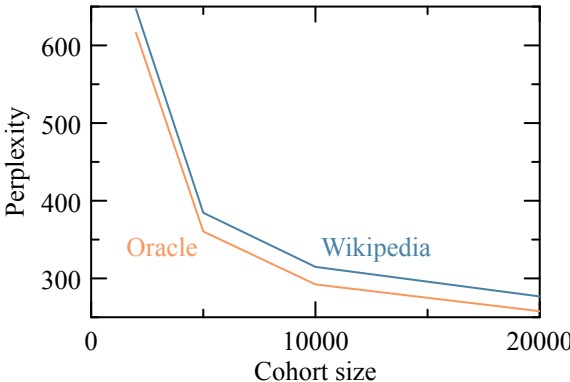

Figure 6: Perplexity trained with different cohort sizes.

## A  Impact of hyperparameters

This section examines different hyperparameters.

### A.1  Experimental design

First, consider the choice to train the public tokenizer on Wikipedia data. To examine the effect of using a more conversational style corpus. To do this, Table 3 takes a subset of the numbers from Table 2 and adds a scenario where a tokenizer on StackOverflow data is used with Reddit data and vice versa. The cross-dataset numbers are highlighted bold in the table.

First, in terms of the KL divergence the StackOverflow data seems a slightly better model for the Reddit distribution than the Wikipedia data is. However, when using PFL to train on Reddit data, but with a StackOverflow-trained tokenizer, the perplexity deteriorates compared to the Wikipedia-trained tokenizer. Second, the reverse experiment looks a bit better but not hugely better. Though the KL divergence from the StackOverflow data to the Reddit data is significantly better than the KL divergence to the Wikipedia data, some of that advantage disappears in the final trained model.

Table 3: The effect of using the Wikipedia corpus against the results in Table 2.

| $\tau$ | Data | Data KLD | LM perp. |
|---|---|---|---|
| *Reddit* | | | |
| BPE | Wikipedia | 0.7826 | 276.5 |
| BPE | **StackOverflow** | 0.6046 | 283.6 |
| BPE | Reddit | 0 | 256.9 |
| BPE | sample | 0.0212 | 257.7 |
| *StackOverflow* | | | |
| BPE | Wikipedia | 1.0629 | 124.6 |
| BPE | **Reddit** | 0.5315 | 118.8 |
| BPE | StackOverflow | 0 | 108.2 |
| BPE | sample | 0.0089 | 108.7 |

Table 4: The effect of varying the vocabulary size.

| Vocab size | Reddit | | StackOverflow | |
|---|---|---|---|---|
| | Wiki | Oracle | Wiki | Oracle |
| 5,000 | 304.3 | 282.2 | 136.3 | 116.8 |
| 10,000 | 276.5 | 256.9 | 124.6 | 108.2 |
| 50,000 | 243.9 | 225.4 | 111.5 | 101.5 |
| 100,000 | 231.2 | 217.9 | 108.9 | 100.5 |

Then, consider the choice of vocabulary size, here the number of distinct tokens. Table 4 shows the perplexities for the baseline ("Wiki") and ceiling ("oracle") experiments. Though the absolute numbers change, the trends do not change.

Similarly for changing model architectures. This paper has presented results on an LSTM model. Table 5 shows results on a Transformer model. Again, though the absolute numbers change, the trends do not change.

### A.2  Other hyperparameters

We consider two hyperparameter choices for experiments: first, the privacy budget, and secondly, the cohort size.

Figure 5 shows the effect of different privacy

Table 5: The effect of changing model architectures.

| Model architecture | Reddit | | StackOverflow | |
|---|---|---|---|---|
| | Wiki | Oracle | Wiki | Oracle |
| Transformer | 261.9 | 244.8 | 117.4 | 107.0 |
| LSTM | 276.5 | 256.9 | 124.6 | 108.2 |

parameters. The effects are not huge, but clearly differential privacy does impede learning somewhat.

Figure 6 shows the effect of differing cohort sizes. A larger cohort size implies a better signal-to-noise ratio when training with differential privacy. However, for practical reasons it is preferable for cohorts to be smaller. 10,000 is a happy medium between good performance and practicality. Also, again, though the absolute numbers change, the trends do not change.