# OpenReview forum: "Training a Tokenizer for Free with Private Federated Learning"
_aclweb.org/ACL/2022/Workshop/FL4NLP — FL4NLP@ACL2022_

### Official Review · Reviewer_gZjY · 2022-03-21
**The authors propose a method to train a 'matched' tokenizer alongside the decentralized and private federated learning of an NLP model over the client data.**

**Rating:** 5
**Confidence:** 3

**Review:**

The authors propose a method to train a 'matched' tokenizer alongside the decentralized and private federated learning of an NLP model over the client data. In particular, the authors consider the problem of having tokenizer for the NLP model that is reflective of the data on the clients that participate in the decentralized federated learning process. When the tokenizer is not matched with the private client, such as when the tokenizer is trained on a public dataset, the authors demonstrate a significant drop in accuracy of the trained model, compared to when using an oracle tokenizer i.e. when the tokenizer is trained on the client data itself. While having a matched tokenizer is essential, training tokenizer on the private client is quite challenging and can potentially cause additional privacy leakage over the existing leakage from DP based FL. Hence, the authors propose a new protocol that samples new datasets for tokenizer up dation using the language model trained using the DP based FL itself. This additional step is integrated into the existing federated learning protocol, and the authors claim that there is no additional privacy leakage. Experiments with many settings are provided that demonstrate that the proposed schemes can match the language model performance of the federated training with oracle tokenizer.

While the problem considered is interesting and relevant, and the algorithm also has some novelty, the claim that there is no additional privacy leakage is not proved formally. In particular, when the tokenizer is modified during the private federated learning, it essentially splits the training into different stages with their own DP guarantees. I don't think the post-processing guarantee of DP applies in such a scenario. A composition analysis to bound the DP privacy budget is needed.

---

### Official Review · Reviewer_n2wc · 2022-03-24
**Reasonable idea, good results, but more work needs to be done before this can be used in practice**

**Rating:** 7
**Confidence:** 2

**Review:**

Thanks for the submission! I enjoyed reading this paper. The goal of this paper is to improve the tokenizer using the samples matching the real distribution of the data without incurring an additional PFL budget. The basic idea is to start with a tokenized from a public dataset, which might not match real data, and then improve that tokenizer using samples obtained from the trained model. After that, replace the old tokenizer with the new one and repeat this process. Evaluations show good results on Reddit and StackOverflow datasets.

1. To apply it to real-world use cases, it’s a bit unclear when we should start sampling the trained model and using the new tokenizer. Experiments shown in Section 5.4 seem to suggest that there is no easy answer, and it might depend on the underlying dataset and algorithm. Given that, any suggestions on how ML practitioners can adopt this? I assume they cannot try multiple options and pick the best one because that would require an additional budget?
2. The experiments seem to be conducted with a fixed budget. I am curious to learn how the proposed algorithm compares with baselines if we got the chance to increase the budget to hit a target perplexity?
3. The evaluation results look good. I am curious if the improvements can also be proven in theory as well. And is it possible to quantify the improvements before training?
4. Are there any limitations of the proposed algorithm?
5. IIUC, before replacing the old tokenizer with the new one, we will need to pause the current process, and use samples from the model to train the new tokenizer. In practice, how long does it take to bring the new tokenizer to a reasonable state, and will this delay be an issue?
6. Is it viable to allocate some dedicated budget to train the tokenizer, say 20%? Is there any estimation of how the proposed algorithm compares to that?

---

### Official Review · Reviewer_zXFH · 2022-03-24

**Rating:** 7
**Confidence:** 4

**Review:**

This paper provides a novel method on training a tokenizer along with the language model privately in a federated learning setting. By utilizing the post-processing theorem of differential privacy, the authors claim that the proposed method satisfies DP without additional privacy cost on training the tokenizer. Empirical results show that the proposed method outperforms heavy-hitters algorithm both in terms of privacy and utility.

In general this paper is well written, with enough background knowledge explained for readers to understand. The motivation is also clear and the algorithm description makes sense. Here are some comments I have to improve the work:
- The authors should clearly clarify what type of privacy the proposed method is protecting. It seems that client-level privacy is enforced and a trustworthy server is assumed. I feel it is important to explicitly state this so that it is clear where the clipping and noise is happening in the FL algorithm.
- It seems from that the proposed method outperforms heavy hitters algorithm even omitting the extra privacy budget induced by the latter. Could the authors provide the exact \epsilon and \delta for the heavy hitters algorithm? Alternatively, could the authors show the utility performance difference given the same privacy budget, including the separate privacy budget, in order to see how much the proposed method outperforms the former.
- There are two minor questions during training a sub-word tokenizer: 1. How does it encode the word when there are multiple sub word combinations? Does it simply search for the one that appears earliest in the dictionary? 2. When updating model embeddings with sub-words, it doesn't seem to be a bijection: different combinations of subwords could result in the same summation, causing words with different semantic meanings to be mapped to the same embedding. Could the authors explain whether this will cause problem to the proposed method?

---

### Official Review · Reviewer_9nuF · 2022-03-24

**Rating:** 6
**Confidence:** 3

**Review:**

This paper proposes a federated learning framework to train a tokenizer while it does not require additional privacy budget in differential privacy. Training the tokenizer is an important part of learning a language model (e.g., Transformer and BERT), but to my best knowledge, it is the first work to study how to train the tokenizer in federated learning setting.
Although the first two contributions (i.e., 1) performance degradation from training with a different distribution and 2) sub-word tokenizer eliminates the out-of-vocabulary problem) are quite obvious, I appreciate this work and advocate accepting the article in this workshop to discuss further. Here are some of my concerns:
1.	I am confused about the system and privacy model, especially how public (Wiki) and private (Reddit or StackOverflow) dataset is distributed over the server/clients. The authors assume that stale tokenizer is trained with the public dataset with a certain privacy budget. However, if the public dataset is utilized to train the model, why should differential privacy be applied? In addition, who generates dataset by utilizing the stale (or old) tokenizer and who update the model embeddings? Clarification about these questions from the FL perspective can improve the paper.
2.	In experiments, it would be better to highlight the paper’s contribution if comparing the two settings in the same privacy budget: 1) proposed scheme (i.e., train old tokenizer with DP and train the new tokenizer without additional privacy budget) and 2) directly train the tokenizer based on private dataset in private FL with the same privacy budget.

---

### Official Review · Reviewer_qmwZ · 2022-03-25
**Interesting idea and reasonable experiment results**

**Rating:** 8
**Confidence:** 4

**Review:**

This paper proposed an interesting idea to learn the Tokenizer/Vocabulary for federated language models. The previous word-level tokenizer/vocabulary can be potentially generated by the following methods: (1) use a public dataset, which may have distribution shift compared to the targeting task; (2) directly collect from target task, which may cause privacy concern; (3) use private heavy hitter to directly collect from target task, which does not seem to provide desirable privacy utility tradeoff. This paper proposed to train a sub-word language model with differentially private federated learning from the targeting task, and then use the trained model to generate/sample words to build the word-level tokenizer. Experiments on stackoverflow, reddits with wiki data as extra public dataset show the effectiveness  of the proposed method.

In general, I think the idea is interesting. The paper is well written, technically solid, and the experiments seem to make sense. I think the draft can be further improved by clarifying the following
(1) Why do we still want to sample a word-level tokenizer if we can train good models with sub-word tokenizer?
(2) I cannot get the intuition why the proposed method can be better than private heavy hitters. Could the authors provide more intuition and highlight it in experiments?

The authors may also be interested in the following paper and blogpost that show how to get DP in FL in practice:
Practical and Private (Deep) Learning without Sampling or Shuffling https://arxiv.org/abs/2103.00039
Federated Learning with Formal Differential Privacy Guarantees https://ai.googleblog.com/2022/02/federated-learning-with-formal.html

---

### Decision · Program_Chairs · 2022-03-26

Accept